# Antibiotic Resistance and Biofilm Formation in *Enterococcus* spp. Isolated from Urinary Tract Infections

**DOI:** 10.3390/pathogens12010034

**Published:** 2022-12-25

**Authors:** Maha A. Khalil, Jamal A. Alorabi, Lamya M. Al-Otaibi, Sameh S. Ali, Sobhy E. Elsilk

**Affiliations:** 1Biology Department, College of Science, Taif University, P.O. Box 11099, Taif 21944, Saudi Arabia; 2Botany and Microbiology Department, Faculty of Science, Tanta University, Tanta 31527, Egypt; 3Biofuels Institute, School of the Environment and Safety Engineering, Jiangsu University, Zhenjiang 212013, China

**Keywords:** *Enterococci*, antibiotic resistance, biofilm formation, virulence genes, PCR

## Abstract

Background: A urinary tract infection (UTI) resulting from multidrug-resistant (MDR) enterococci is a common disease with few therapeutic options. About 15% of urinary tract infections are caused by biofilm-producing *Enterococcus* spp. Therefore, the objective of this study was to identify the MDR enterococci associated with UTIs and assess their potential to produce biofilms. Methods: Thirty *Enterococcus* isolates were obtained from urine samples collected from UTI patients at King Abdulaziz Specialist Hospital in Taif, Saudi Arabia. The antimicrobial resistance profiles of the isolates were evaluated using disk diffusion techniques against 15 antimicrobial agents. Two techniques, Congo red agar (CRA) and a microtiter plate (MTP), were used to assess the potential of the isolates to produce biofilms. The enterococcal isolates were screened for biofilm-related genes, *esp*; *ebpA*; and *ebpB*, using the PCR method. Results: The molecular identification of the collected bacteria revealed the presence of 73.3% *Enterococcus faecalis* and 26.6% *Enterococcus faecium*. The antibiotic susceptibility test revealed that all the tested *Enterococcus* spp. were resistant to all antimicrobials except for linezolid and tigecycline. Additionally, by employing the CRA and MTP techniques, 76.6% and 100% of the *Enterococcus* isolates were able to generate biofilms, respectively. In terms of the association between the antibiotic resistance and biofilm’s formation, it was observed that isolates capable of creating strong biofilms were extremely resistant to most of the antibiotics tested. The obtained data showed that all the tested isolates had biofilm-encoding genes. Conclusions: Our research revealed that the biofilm-producing enterococci bacteria that causes urinary tract infections were resistant to antibiotics. Therefore, it is necessary to seek other pharmacological treatments if antibiotic medicine fails.

## 1. Introduction

The colonization of any urinary tract tissue, from the renal cortex to the urethral meatus, is described as a urinary tract infection (UTI) [1]. A UTI is one of the most prevalent nosocomial and community-acquired illnesses in humans [2]. *Escherichia coli*, *Enterococcus* sp., *Klebsiella* sp., *Staphylococcus saprophyticus*, *Pseudomonas aeruginosa*, *Staphylococcus aureus*, and *Proteus* sp. are the most common bacteria that cause UTIs [3]. Enterococci are an opportunistic pathogen that may cause severe and potentially serious infections in immunocompromised individuals, as well as those receiving broad-spectrum antibiotic treatment or those requiring extended hospitalization [4]. *Enterococcus faecalis* or *Enterococcus faecium* are the most common cause of UTIs and account for around 95% of human enterococcal infections, followed by *E. coli* [5]. In Saudi Arabia, the most often occurring bacteria that cause a UTI are *E. coli*, *Enterococcus* sp., and *Klebsiella* sp. [6].

Enterococci are attracting attention due to their multidrug resistance, which explains their preponderance in infections [7,8,9,10]. They are innately resistant to a broad spectrum of routinely used antibiotics, including cephalosporins, aminoglycosides, macrolides, and trimethoprim-sulfamethoxazole [11,12]. Bacteria that form biofilms cause more than 65% of nosocomial infections and 80% of bacterial infections, which may become a severe concern in the area of urology [13]. The ability of *Enterococcus* to produce biofilms is one of its most noticeable pathogenicity characteristics. This property enables the colonization of inert and biological surfaces as well as mediates the adherence to host cells. These biofilms are communities of cells that have been permanently attached to a variety of biotic and abiotic surfaces and are encased in a hydrated matrix of exopolymeric molecules, proteins, polysaccharides, and nucleic acids [14]. There is a noticeable variation in the behavior of bacteria in biofilms compared to their free-living counterparts. Biofilms protect their members from host immunological responses, phagocytosis, and antibiotics [15], thereby complicating the disease. While it is widely agreed that antibiotic therapy is the most important and successful method currently available for managing microbial infections, antibiotic therapy alone is sometimes insufficient to eradicate the bacterial biofilm and the associated infections [16,17].

Numerous enterococci virulence factors have been identified as being associated with the biofilm’s formation, including adhesions and generated virulence factors. *Asa 1* (aggregation substance), *Esp* (extracellular surface protein), and *Ebp* (endocarditis and biofilm-associated pili) are the most important adhesion factors [18]. *CylA* (cytolysin), *GelEA* (gelatinase), and *Hyl* (hyaluronidase) are the produced pathogenic components of enterococci and play a role in pathogenesis [19]. *Esp* is a cell wall-associated protein that has been involved in the colonization, persistence, and production of biofilms of bacteria in the urinary system [20]. *Asa1* is a pheromone-inducible protein that acts as a virulence factor by promoting a bacterial adhesion to renal tubular cells. Additionally, the *Ebp* operon contains the genes *ebpA*, *ebpB*, and *ebpC* which are involved in the creation of pills by enterococci and are required for bacteria to cause a UTI [21]. The recent focus has centered on the study of the genes implicated in the development of a biofilm and their significance in infections caused by enterococci [17].

*Enterococcus faecalis* and *Enterococcus faecium* have emerged as particularly important etiological pathogens of nosocomial infections, notably urinary tract infections. Therefore, the goal of this study was to investigate the MDR and biofilm formation among enterococci isolated from urinary tract infections and to assess the relationship between the two phenomena.

## 2. Materials and Methods

### 2.1. Isolates Selection

The clinicians of the King Abdulaziz Specialist Hospital in Taif, Saudi Arabia examined 224 patients with UTI symptoms for the isolation of enterococcal bacteria. The clinicians followed the guidelines and the standard protocols that are compatible with the requirements of the Declaration of Helsinki [22]. The study’s protocol was approved by the review board of the King Abdulaziz Specialist Hospital, Taif, KSA (SRU-KAASH-107) and the samples were collected between October 2018 and October 2019. There were 121 females (54.01%) and 103 males (45.98%), aged 6 to 112 years. The collected specimens were transferred to the Laboratory of Microbiology at the Department of Biology, the Faculty of Science, Taif University (the female section) for a further testing of the experiments. 

### 2.2. Enterococcal Isolation 

All urine samples were microscopically inspected after 10 min of centrifugation at 1000 rpm. Pus cells, RBCs, epithelial cells, casts, and crystals were found as signs of a urine infection. Using a sterile standard loop, all urine samples were cultured on conventional culture media, including blood agar (Oxoid, Basingstoke, UK), and Cystine–Lactose–Electrolyte Deficient (CLED) agar (Oxoid, Basingstoke, UK). The plates were incubated for 24–48 h at 37 °C. Gram’s stain; catalase, motility, and pigment tests; *Enterococcus* selective media (Bile-esculin-azide agar and Slanetzand Bartley agar); and a salt tolerance test (6.5% NaCl) was used to identify *Enterococcus* sp. Such brown-to-black colonies in the surrounding environment were suspected to be *Enterococcus*. Other testing, including growth in very alkaline conditions (pH = 9.6) and high temperatures, showed the isolates to be enterococci. An API 20 Strep (BioMerieux, St. Louis, MO, USA) biochemical test kit was also used to validate the enterococcal isolates biochemically.

### 2.3. Antibiotic Susceptibility Test

Kirby Bauer’s disc diffusion method was used to investigate the antibiotic sensitivity of the antimicrobial agents against clinical enterococcal isolates [23]. *E. faecalis* ATCC 2921 was used as a reference strain for the antibiotic susceptibility tests. Antimicrobials (MAST Diagnostics, Liverpool, UK) were categorized by classes and abbreviation, with concentrations, as shown in Appendix A. A 1 × 10^5^ CFU/mL, 20 μL overnight enterococci culture was added to 100 mL of Nutritional Broth (NB) medium and incubated for 24 h at 37 °C/120 rpm. A bacterial culture (100 μL) was streaked onto Mueller Hinton Agar (MHA; Oxoid, Ltd.) plates using cotton swabs and incubated aerobically at 37 °C for 18–24 h. The inhibitory zones (mm) were measured after the incubation, and the findings were classed as sensitive or resistant using the CLSI interpretation criteria [23]. The ‘Intermediate’ results were categorized as ‘resistant’. Multidrug-resistant (MDR) refers to bacteria that have developed a resistance to at least one agent in three or more antimicrobial classes, as described by Magiorakos et al. [24].

### 2.4. Enterococcal Biofilm Formation

The Congo red agar (CRA) and microtiter plate technique (MTP) methods were used to examine the biofilm’s formation by MDR enterococcal isolates [24].

#### 2.4.1. Using Congo Red Agar (CRA)

The formation of the biofilm was assessed qualitatively by culturing enterococcal isolates on CRA plates [24,25]. To prepare CRA plates, 1 L of brain heart infusion agar (Merck, Darmstadt, Germany) was mixed with 0.8 g of Congo red dye (Merck, Darmstadt, Germany), and 36 g of saccharose. The plates were incubated for 24 h at 37 °C and followed overnight at room temperature. Black colonies were assessed to be strains with a high capacity to produce a biofilm (P; producer), whereas strains with red colonies were identified as the strains incapable of producing the biofilm (NP; non-producer).

#### 2.4.2. Using Microtiter Plates (MTP)

The enterococci biofilm production was assessed using a semi-quantitative adhesion test on MTP. Briefly, the bacterial isolates were grown overnight at 37 °C in trypticase-soy broth (TSB, Merck, Germany). The enterococcal cultures were diluted 1:100 in fresh TSB containing 2% glucose. The diluted solution was then applied to the wells of a flat-bottomed polystyrene microtiter plate (Sigma Aldrich, Darmstadt, Germany) and incubated at 37 °C for 24 h. Wells with sterile TSB containing 2% glucose served as the negative controls. After incubation, each well was drained and cleaned three times with 300 μL of phosphate-buffered saline (PBS; pH 7.2). The adhesive bacteria were fixed in 95% ethanol for 5 min and stained with 100 μL of 1% crystal violet. This was followed by three thorough cleanings with 300 μL of sterile distilled water. The microplates were then dried in the air. The optical density (OD) of the stained adherent was measured at 570 nm using an ELISA reader (BioTek Instruments, Winooski, VT, USA). The strains were divided into groups based on the OD values produced by the bacterial biofilms, as described by Christensen et al. [26], with minor modifications, as indicated by Stepanovic et al. [27]. The bacterial strains were classified as follow: the OD value < 0.120 as weak/negative biofilm-producing (WP/NP) isolates, and those with OD > 0.120 and <0.240 were regarded as moderate biofilm producers (MP). An OD value > 0.240 was indicative of high biofilm-producing bacterial (HP) strains.

### 2.5. PCR Amplification and Phylogenetic Analysis

The genomes of the bacterial strains were isolated from pure bacterial cultures and sequenced based on the manufacturer’s instructions for the DNA extraction kit (Promega, USA). Two universal primers, 1492R (5′-TACGGYTACCTTGTTCGACTT-3′) and 27F (5′-AGAGTTTGATCTGGCTCAG-3′) were used in the 16S rRNA gene amplification. The Bio-Rad S1000™ Thermal Cycler (Bio-Rad, Hercules, CA, USA) was used to perform the PCR amplification [27]. In a TBE buffer 1×, an agarose gel at a concentration of 1.5% (*w*/*v*) was utilized. The gels were stained with ethidium bromide (EtBr) solution at a concentration of 0.5 mg/mL. A UV transilluminator (Biotium, Fremont, CA, USA) was used to visualize the DNA, and the results were photographed and reported. Purified PCR products were sequenced at Macrogen Co., South Korea. The initial sequence analysis was performed using BlastN (http://www.ncbi.nlm.nih.gov/BLAST/ accessed on 20 September 2020). A neighbor-joining phylogenetic tree was constructed based on 16S rRNA gene sequences of enterococci members and the related genera using molecular evolutionary genetics analysis version 7 (MEGA7) software [28]. 

### 2.6. Detection of Biofilm-Associated Genes

The presence of the biofilm-associated genes was also detected in this study using a PCR. Specific primers were used for the PCR testing of all the bacterial isolates to detect virulence genes encoding *esp*, *ebpA*, and *ebpB* genes. The primers used for the determination of these virulence genes were previously described [29,30,31]. The PCR procedure for all the genes was as follows: initial denaturation at 95 °C for 5 min, followed by 30 cycles of 30 s; denaturation at 94 °C, 30 s; primer annealing at 58 °C; and a 1 min extension at 72 °C. Following these cycles, a final 7 min extension step at 72 °C was performed. Gel electrophoresis (Cleaver Scientific, Warwickshire, UK), utilizing a 1.5% agarose gel (Cleaver Scientific Ltd., Warwickshire, UK), was used to analyze the amplification products.

### 2.7. Statistical Analysis

All the obtained data were analyzed by GraphPad Prism (GraphPad Software, Inc., San Diego, CA, USA) version (8.0.2) using a paired t-test to show the variations between the resistance and biofilm-forming ability of the investigated *Enterococcus* sp. The obtained results were significant at * *p* ≤ 0.05, ** *p* ≤ 0.01, *** *p* ≤ 0.001, and **** *p* ≤ 0.0001, while *p* > 0.05 is non-significant (ns).

## 3. Results and Discussion

Enterococci may be responsible for 15% of urinary tract infections in critical care units (ICU) [18]. In this study, only 30 (13%) enterococci bacterial isolates were found in 224 patient urine samples. Female culture-positive urine specimens were reported more frequently than male culture-positive urine specimens. Females have a greater incidence of urinary tract infections due to anatomical differences and a reliance on the male urinary system’s natural defensive systems [32]. The morphological and biochemical tests confirmed that all isolates (*n* = 30) were *Enterococcus* species and designated as E01–E30. 

The susceptibility and resistance rates among enterococcal isolates were shown in Figure 1. According to the data shown in Figure 1A, the majority of *E. faecalis* strains were resistant to tetracycline (86.36%), erythromycin (81.81%), levofloxacin, and quinpristin–dalfopristin (77.27%). There were less resistant isolates, such as gentamicin (50%), vancomycin (31%), and streptomycin (27%). Nitrofurantoin showed the lowest degree of resistance (18.18%) regarding the *E. faecium* strains; the maximum resistance rate (100%) was seen for tetracycline and erythromycin, while the lowest resistance rate (12.5%) was observed for streptomycin (Figure 1B). All the *Enterococcus* isolate was completely susceptible to linezolid and tigecycline. According to a statistical analysis using the paired *t*-test, the resistance of *E. faecium* against the investigated antimicrobial agents was insignificantly (*p* = 0.8450) higher than *E. faecalis,* as shown in Figure 1C. A resistance to fluoroquinolones (levofloxacin) may be the result of alterations in the target enzymes related to the DNA replication, hence preventing the bacterial proliferation [33]. Chow [34] stated that all the enterococci tested exhibited a modest inherent resistance to aminoglycosides, including gentamicin, which was regarded as the most often recommended aminoglycoside for enterococci infections. Comparable observations by Maschieto et al. [35], Rajendiran et al. [36], and Wei et al. [37] revealed a moderate level of resistance to streptomycin. 

MDR *Enterococcus* sp. strains displayed five resistance profiles for different classes of tested antimicrobials, as shown in Table 1, with 74% of isolates resistant to more than four classes. E18, E21, and E14 were the most resistant to seven antimicrobial classes, whereas E05 and E29 showed a resistance to three. These findings corroborated those of Fered et al. [38]. 

Additionally, biofilms played a crucial role in the high incidence, recurrence, and severity of UTIs, causing both acute and chronic infections [39]. As a result, the following tests subjectively recognized the formation of the biofilm in the tested enterococci isolates on a CRA media and quantified it using the MTP method. To summarize the CRA data, biofilm producers (P) appeared as black colonies, whereas non-biofilm producers (NP) appeared as pink colonies (Figure 2). The percentage of isolates forming biofilms was 17 (77.2%) for *E. faecalis* and 6 (75%) for *E. faecium*. Baldassarri et al. [40] observed that 80% of *E. faecalis* and 48% of *E. faecium* were significantly associated with the biofilm production in infected patients. Additionally, Dupre et al. [41] reported that *E. faecalis* produced biofilms to varying degrees in 86.7% of clinical specimens, whereas only 15.6% of *E. faecium* specimens formed biofilms. As illustrated in Figure 3, a biofilm formation was detected using the MTP methods. According to the obtained results, 45.45% of *E. faecalis* isolates were found to be high producers (HP), while 54.54% were found to be moderately adherent (MP) (Figure 3A). Where 37.5% of *E. faecium* isolates were found to be highly produced, 62.5% were found to be moderate producers (Figure 3B). As shown in Figure 4, a comparison of the two biofilm detection methods demonstrated that the biofilm’s development was subjectively seen in CRA, although only 23 (76.6%) of the isolates achieved the biofilm production criteria (Figure 4A). Using the MTP test, all 30 of the investigated isolates produced biofilms with varying degrees (Figure 4B). The findings of two techniques for determining the production of biofilms by the *Enterococcus* species were comparable, although it seems that MTP is the more precise and dependable methodology [24,41,42]. Lasa [43] obtained comparable findings; Mathur et al. [44] likewise encouraged the adoption of the MTP approach because of its excellent specificity, sensitivity, and positive predictive value. However, Hassan et al. [45] and Kafil et al. [18] advocated for the use of CRA to identify biofilms.

Statistical analysis indicated a significant correlation between the *Enterococcus* species that form biofilms and the antibiotic resistance to certain antimicrobials (Figure 5). A resistance to several antibiotics, including penicillin, ampicillin, gentamicin, nitrofurantoin, levofloxacin, and erythromycin, was significantly higher in high biofilm producers than in moderate biofilm producers (Figure 5A). According to the obtained data from the statistical analysis using *t*-test analysis, as shown in Figure 5B, a penicillin resistance (56.2% vs. 43.7%, *p =* 0.0002), ampicillin resistance (54.5% vs. 45.4%, *p* = 0.0008), gentamicin resistance (53.8% vs. 46.1%, *p* = 0.0006), nitrofurantoin resistance (66.6% vs. 33.3%, *p* < 0.0001), and levofloxacin and erythromycin resistance (52.1% vs. 47.8%, *p* = 0.0054). Similarly, various studies have shown a significant correlation between the formation of the biofilm and antibiotic resistance in the enterococci isolated from urinary tract infections [45,46,47], whereas a norfloxacin, ciprofloxacin, quinupristin/dalfopristin, and tetracycline resistance were rather uncommon among biofilm-forming bacteria. As a result, these antibiotics may be utilized effectively to treat UTIs caused by enterococcal isolates from biofilm producers. Akhter et al. [46] found a strong correlation between the biofilm’s formation in *Enterococcus* spp. isolated from urinary tract infections and the antibiotic resistance to amoxicillin, co-trimoxazole, ciprofloxacin, gentamycin, cefotaxime, and cefuroxime. Additionally, Fallah et al. [17] reported that the most effective drugs against the *Enterococcus* species were linezolid, chloramphenicol, and nitrofurantoin.

As depicted in Figure 6, 27 isolates were identified as *E. faecalis* and 3 as *E. faecium*. The 16S rRNA gene sequences may be used to place diagnoses into a phylogenetic tree and can be connected to hundreds of database sequences [48]. The comparison of these sequences enables the differentiation of organisms at the genus level across all of the major bacterial phyla, as well as the categorization of strains at numerous levels, including what is currently referred to as the species and subspecies levels [49].

Numerous investigations have been conducted on the enterococci virulence genes involved in the biofilm’s formation. However, the pathogenic mechanisms and genes involved in the biofilm’s formation in enterococci were deemed to be insufficient [18]. In this study, the presence of biofilm-related genes was evaluated in *E. facealis* and *E. faecium* by PCR. The findings shown in Figure 7 indicated that the *Esp*, *EbpA*, and *EbpB* genes were detected in all of the tested strains at a 100% prevalence rate. Shankar et al. [50] detected *Esp*, a cell wall-related protein, in the enterococci strains. It has been shown that it promotes the adhesion, colonization, and evasion of the immune system, as well as having an influence on antibiotic resistance [51,52]. Soares et al. [53] established that *Enterococcus faecalis* harbored *GelE, Esp*, and *Asa 1* genes. In clinical UTI isolates, Kafil and Mobarez [18] found a relationship between the virulence factors and the biofilm’s development. The findings of the statistical analysis indicate that there was a significant correlation between the frequency of *esp, ebpA*, and *ebpB* genes with a biofilm development in all of the tested isolates (*p* < 0.05). All of the investigated genes were found in high biofilm producers (42%) and intermediate biofilm producers (57%).

## 4. Conclusions 

The results of this research reveal that the emergence of an antibiotic resistance in enterococci that cause urinary tract infections limits the therapeutic use of antimicrobials. In addition, the spread of MDR enterococci isolates poses a threat to hospitalized patients’ health. The research showed that all isolates had the potential to form a biofilm, which complicates their treatment with antibiotics, confirming the critical need to develop novel antimicrobial agents that control the infection associated with the development of a biofilm. 

## Figures and Tables

**Figure 1 pathogens-12-00034-f001:**
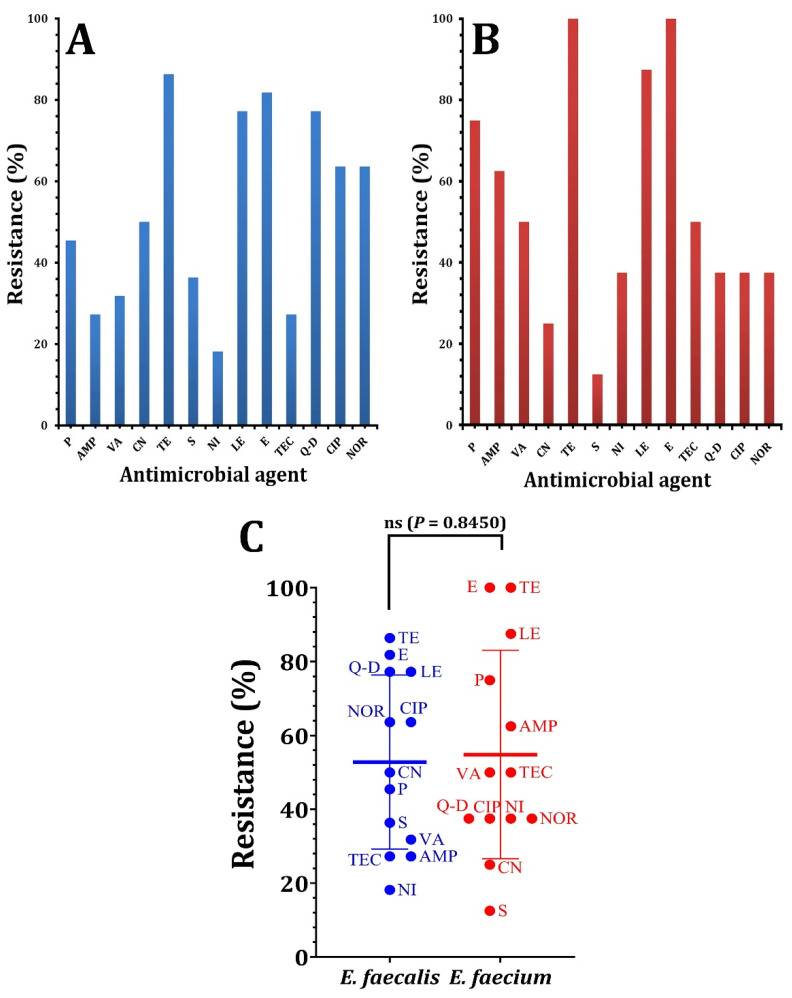
The percentages of antibiotic susceptibility of *Enterococcus* species against antimicrobial agents. (**A**) antibiotic susceptibility of *E. feacalis*, (**B**) antibiotic susceptibility of *E. faecium*. (**C**) the statistical analysis using paired t-test comparing the resistance of *Enterococcus* species against antimicrobial agents. *p* > 0.05 is non-significant (ns).

**Figure 2 pathogens-12-00034-f002:**
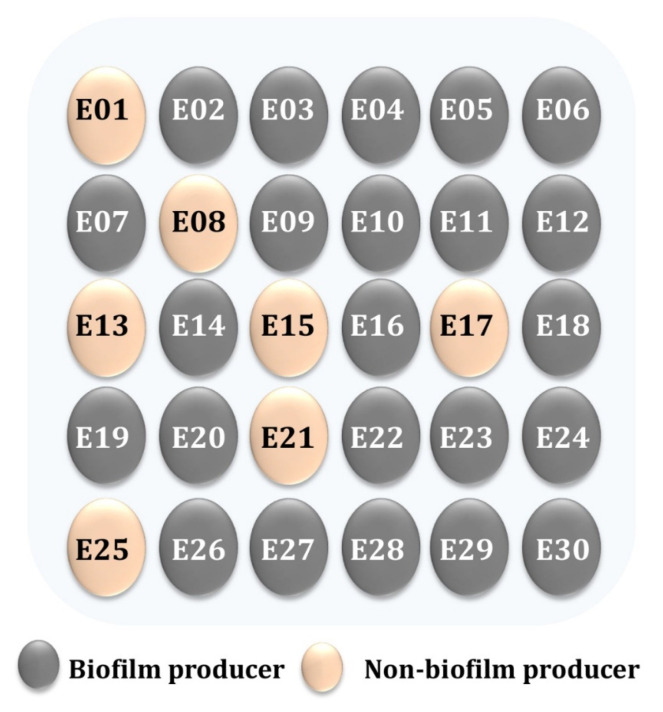
Biofilm production by tested *Enterococcus* species on CRA media.

**Figure 3 pathogens-12-00034-f003:**
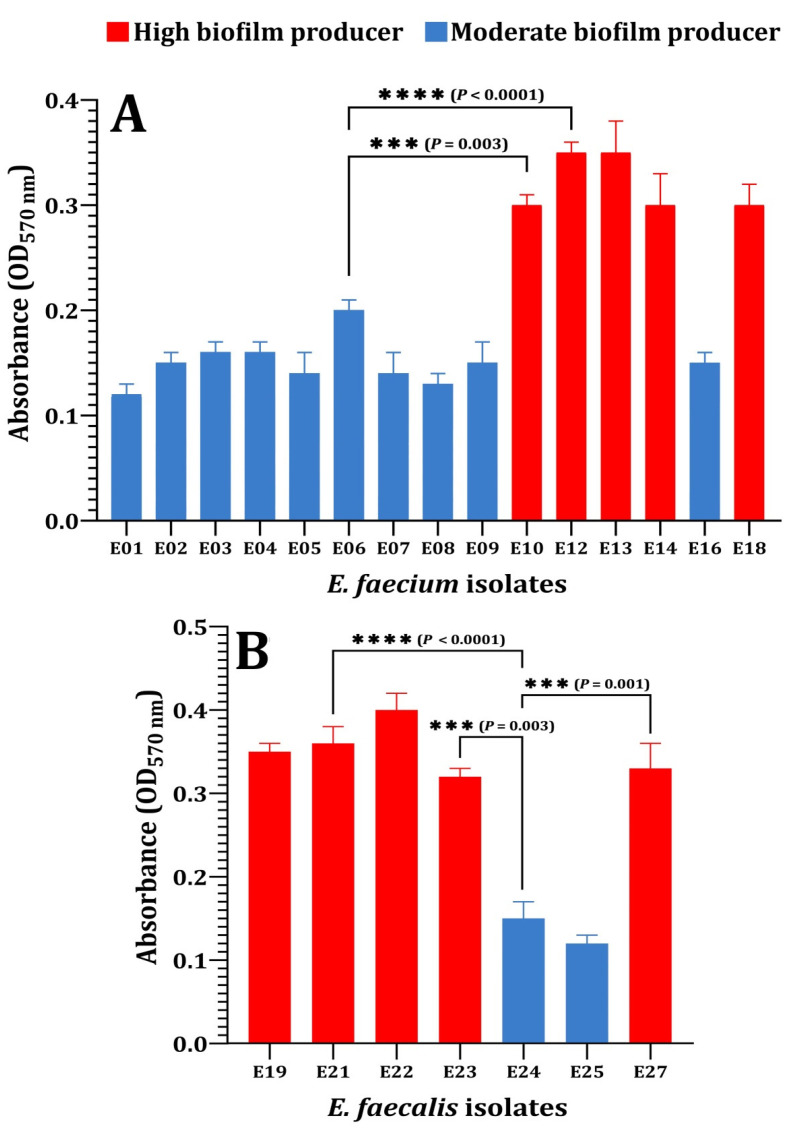
Biofilm production by *Enterococcus* species using the MTP method. (**A**) *E. faecium*, (**B**) *E. faecalis*. Biofilm formation was determined using the crystal violet assay at optical density at 570 nm (OD_570_). High biofilm producer; OD value > 0.240, moderate biofilm producer; OD value > 0.120 and <0.240. *p*-values for significantly different mean values, *p* < 0.001 (***), *p* < 0.0001 (****).

**Figure 4 pathogens-12-00034-f004:**
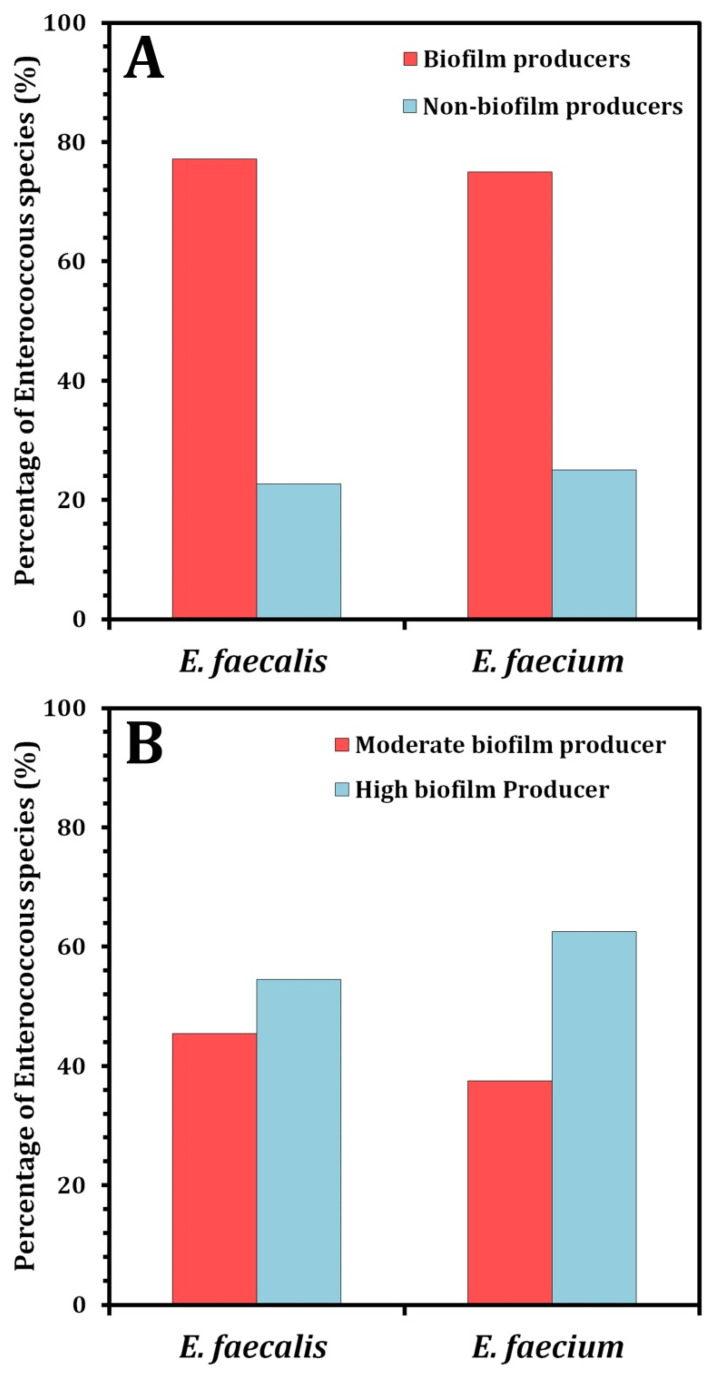
Comparing biofilm forming ability of *Enterococcus* species in CRA and MTP techniques. (**A**) Biofilm formation using CRA method. (**B**) Biofilm formation using MTP assay.

**Figure 5 pathogens-12-00034-f005:**
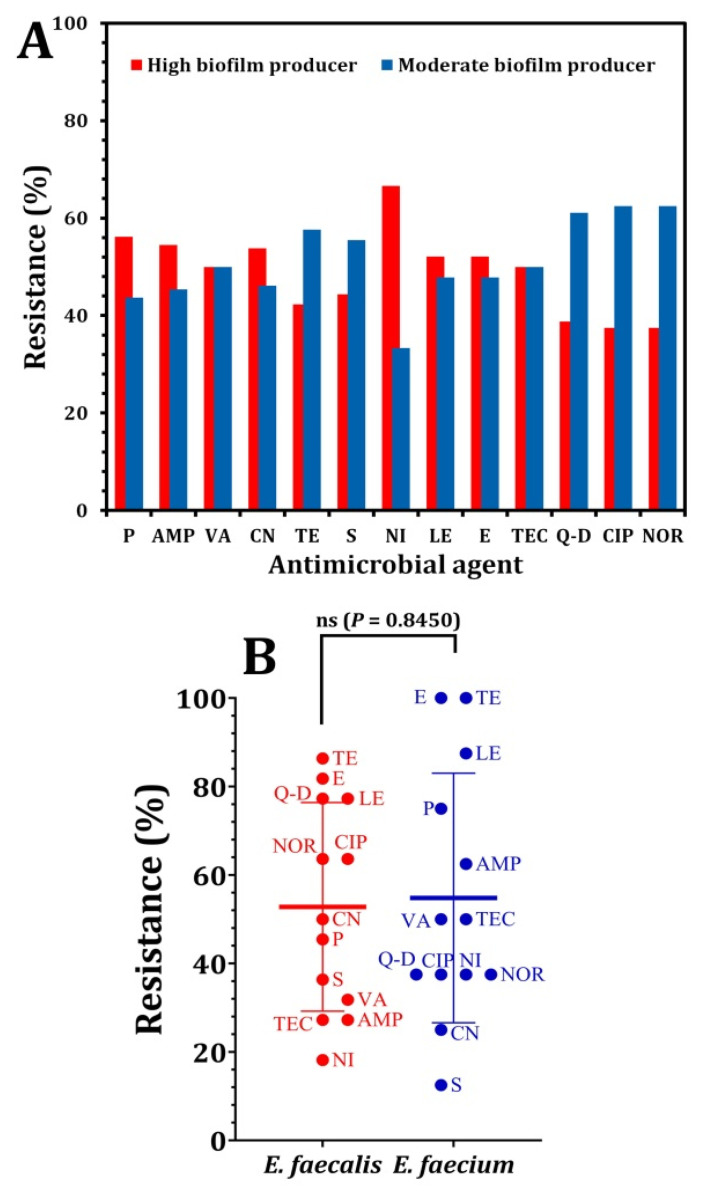
Correlation between the production of biofilms and the pattern of antibiotic resistance in enterococci isolates (**A**) and the paired *t*-test analysis (**B**). *p* > 0.05 is non-significant (ns).

**Figure 6 pathogens-12-00034-f006:**
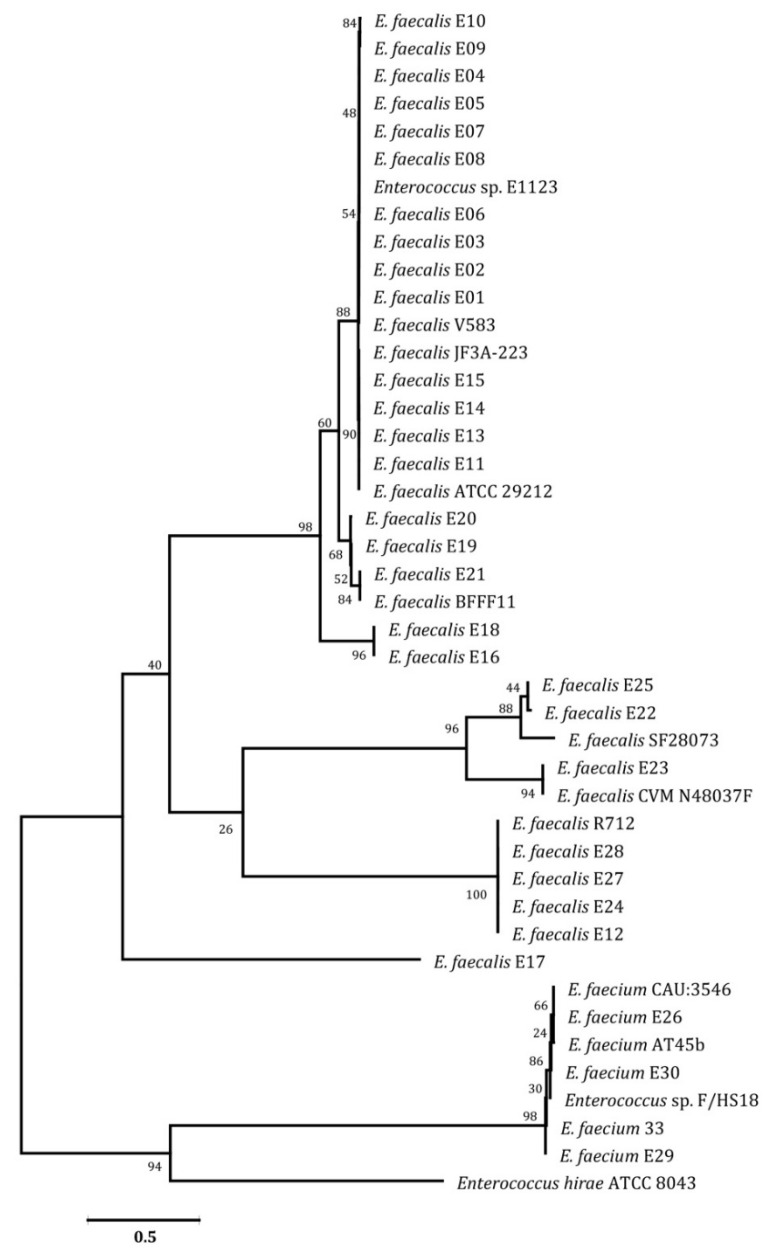
A neighbor-joining tree based on 16S rRNA gene sequences of enterococci strains and related genera. The scale bar indicates 0.05 substitutions per nucleotide position.

**Figure 7 pathogens-12-00034-f007:**
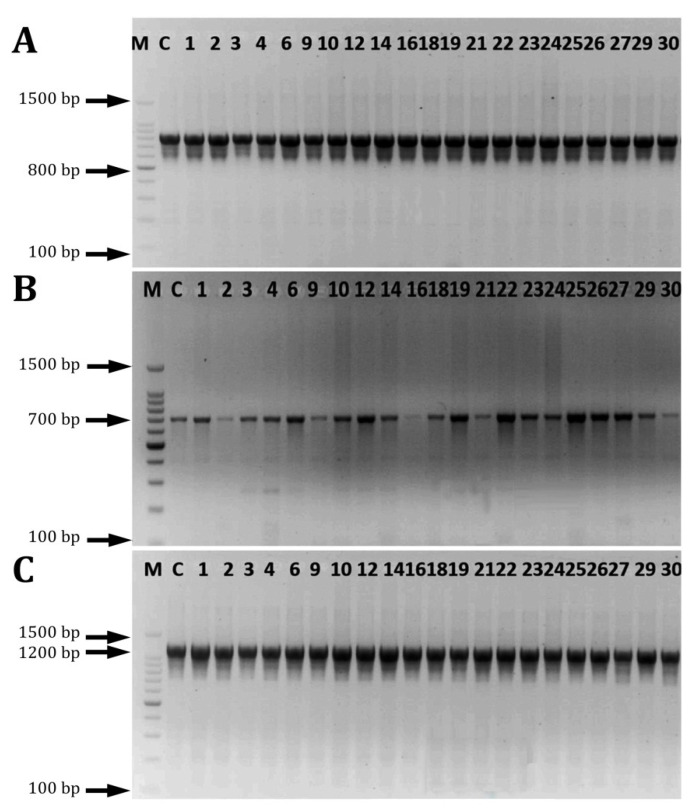
PCR detection of biofilm-related genes in *Enterococcus* isolates. (**A**) *esp* gene, (**B**) *ebpA* gene, (**C**) *ebpB* gene, M molecular weight markers (100 bp), and C control (reference strain).

**Table 1 pathogens-12-00034-t001:** Resistance patterns of MDR *Enterococcus* sp. isolates against different antimicrobial agents.

Pattern Code	Isolate Code	^#^ Antimicrobial Resistance	No. of Antimicrobial Classes
**A**	E05	TE ^V^, CIP ^VII^, NOR ^VII^, Q-D ^IX^	3
	E29	P ^I^, AMP ^I^, TE ^V^, LE ^VII^	
**B**	E03	VA ^II^, TEC ^II^, TE ^V^, LE ^VII^, CIP ^VII^, NOR ^VII^, E ^VIII^	4
	E11	P ^I^, AMP ^I^, TE ^V^, LE ^VII^, E ^VIII^	
	E13	CN ^III^, LE ^VII^, CIP ^VII^, NOR ^VII^, E ^VIII^, Q-D ^IX^	
	E19	TE ^V^, LE ^VII^, CIP ^VII^, NOR ^VII^, E ^VIII^, Q-D ^IX^	
	E23	P ^I^, AMP ^I^, S ^III^, LE ^VII^, E ^VIII^	
**C**	E01	P ^I^, AMP ^I^, VA ^II^, TEC ^II^, TE ^V^, LE ^VII^, CIP ^VII^, NOR ^VII^, E ^VIII^	5
	E02	CN ^III^, S ^III^, TE ^V^, CIP ^VII^, NOR ^VII^, E ^VIII^, Q-D ^IX^	
	E04, E09	P ^I^, CN ^III^, S ^III^, TE ^V^, CIP ^VII^, NOR ^VII^, LE ^VII^, Q-D ^IX^	
	E06	CN ^III^, S ^III^, TE ^V^, LE ^VII^, CIP ^VII^, NOR ^VII^, E ^VIII^, Q-D	
	E08	AMP ^I^, TE ^V^, LE ^VII^, E ^VIII^, Q-D ^IX^	
	E10	P ^I^, AMP ^I^, VA ^II^, TEC ^II^, TE ^V^, NI^VI^, LE ^VII^	
	E12, E16	CN ^III^, TE ^V^, LE ^VII^, CIP ^VII^, NOR ^VII^, E ^VIII^, Q-D ^IX^	
	E15	P ^I^, AMP ^I^, CN ^III^, TE ^V^, LE ^VII^, E ^VIII^	
	E17	P ^I^, AMP ^I^, VA ^II^,TEC ^II^,TE ^V^, CIP ^VII^, NOR ^VII^,E ^VIII^	
	E20	VA ^II^, TEC ^II^, TE ^V^, CIP ^VII^, NOR ^VII^, E ^VIII^, Q-D ^IX^	
	E26	VA ^II^, TEC ^II^, TE ^V^, NI ^VI^, LE ^VII^, E ^VIII^	
	E30	P ^I^, CN ^III^, TE ^V^, LE ^VII^, Q-D ^IX^	
**D**	E22	P ^I^, CN ^III^, TE ^V^, LE ^VII^, E ^VIII^, Q-D ^IX^	6
	E27	P ^I^, AMP ^I^, VA ^II^, TEC ^II^, CN ^III^, TE ^V^, LE ^VII^, E ^VIII^	
	E28	P ^I^, AMP ^I^, VA ^II^, TEC ^II^, S ^III^,TE ^V^, LE ^VII^, CIP ^VII^, NOR ^VII^, E ^VIII^	
**E**	E18	P ^I^, CN ^III^, S ^III^, TE ^V^, LE ^VII^, CIP ^VII^, NOR ^VII^, E ^VIII^, Q-D ^IX^	7
	E21	P ^I^, VA ^II^, TEC ^II^, CN ^III^, S ^III^, TE ^V^, LE ^VII^, E ^VIII^, Q-D ^IX^	
	E14	P ^I^, AMP ^I^, VA ^II^, TEC ^II^ S ^III^, TE ^V^, NI ^VI^, LE ^VII^, CIP ^VII^, NOR ^VII^, E ^VIII^	

^#^ P: penicillin, AMP: ampicillin, VA: vancomycin, TEC: teicoplanin, CN: gentamicin, S: streptomycin, TE: tetracycline, NI: nitrofurantoin, LE: levofloxacin, CIP: ciprofloxacin, NOR: norfloxacin, E: erythromycin, and Q-D: quinpristin–dalfopristin ^I^ β-Lactam antibiotics, ^II^ glycopeptides, ^III^ aminoglycosides, ^IV^ oxazolidinones, ^V^ tetracyclines, ^VI^ nitrofurantions, ^VII^ fluoroquinolones, ^VIII^ macrolides, ^IX^ streptogramins.

## Data Availability

All data generated or analyzed during this study are included in this published article (and its Appendix A).

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
