# Peer review of "Antibiotic Resistance and Biofilm Formation in Enterococcus spp. Isolated from Urinary Tract Infections"

_pathogens, 2022, doi:10.3390/pathogens12010034_

Round 1

Reviewer 1 Report

The Manuscript submitted to review entitled “Screening and characterization of multidrug-resistant and biofilm forming enterococci isolated from urinary tract infections” presents an interesting topic regarding antibiotic resistance of strains from clinical origin.

The research was designed and performed in an appropriate way, I have no objections to the methodology as well as the methods of performing and interpreting the results. The authors wrote the introduction in a synthetic way, clearly formulated the purpose of the research and clearly presented the conclusions. Nevertheless, some reservations are raised by the graphical presentation of the results. Here are my observations on this matter:

Figure 1. According to the Authors, is it reasonable to assign the % value of 150 (resistance, %) to the axis? While the total is 100%, and otherwise the Authors have not identified values ​​greater than 100%.

Figure 1. Are R1 and R2 labeling necessary? Maybe it's better to indicate the names of the strains directly on the chart - E. faecium and E. faecalis in the right places?

Table 1 and Table 2: It seems to me that these tables are analogous - could they not be presented in the form of one table?

Figure 2. It seems to me that this Figure is too convoluted to interpret compared to the information it contains ... Maybe it's better to mark biofilm production as "+" or "-" instead of entering green and red?

Figure 3. Maybe it's better to mark the axis containing percentage values ​​to the value of 100? Maybe the explanation of the abbreviations HP and MP is better to include in the drawing than just in the description?

Figure 4. Here I don't see the need for different colors at all. One color may be used and the full names of the abbreviations P, NP, HP and MP may be given in the appropriate places above the bars.

Figure 5. Geometric markings on the bars are incomprehensible to me.

Figure 7. The Figure is of low quality - the image is stretched, it looks very unprofessional. Especially the subtitles.

I consider the Manuscript valuable and worthy of publication, especially in the current situation of the antibiotic crisis, but I recommend correcting the Figures and Tables before publication in Pathogens.

Author Response

Kindly find the attached response letter.

Reviewer 2 Report

In this study, the authors did a great job, detecting multi-drug resistant and biofilm-forming enterococci from human urine samples. Their presence in humans poses a serious public health threat. I believe that the findings from this research will help in taking initiative for the selection of proper antibiotics to treat enterococcal infections. However, I have several comments that must be addressed before the further decision. Please find and check my comments below:

Overall comments

·         Please be careful using words throughout the manuscript. “Discovered” and “Detected” are not the same. I believe wording should be improved throughout the manuscript.

·         Please try to minimize the length of the background and conclusion of the abstract.

·         Objective part of your introduction section (the last paragraph of the introduction) must be improved. You must mention the justification and/or knowledge gaps of your study.

·         Objective, materials and methods, and results must be in past form. Please correct them throughout the manuscript.

·         Grammatical errors, especially present form vs past form, must be corrected.

·         You must mention the name of manufacturer, city, and country of any materials you used during your study. I didn’t provide any comments for this in your manuscript. Please mention them properly.

·         The use of streptomycin isn’t appropriate. You should omit any information on streptomycin from your manuscript. But if you want to keep them, please clarify it from your side with proper references.

Comments

The background of the abstract is too long. Please minimize it.

Line 17-18: Please paraphrase this sentence.

Line 15: Please write the short form of multi-drug resistant (MDR) here.

Line 18: “The goal of this study is…” should be “The goal of this study was…”. The objectives always should be in the past form.

Line 18: Please just write “MDR” here.

Line 22: Please use “agents” instead of “drugs” here

Line 22-23: The laboratory investigated the biofilm-forming ability, or you investigated? Please clarify it. If needed, please correct it.

Line 27: “all antibiotics tested were resistant to….”? Antibiotics can’t be resistant to an organism; an organism shows resistance to antibiotics. Please rephrase the sentence.

Line 29: Please write either just “enterococci” or “Enterococcus isolates” here.

Line 29: Please write “respectively” after “….generate biofilms”.

Line 30: Please write “observed” and “strong” instead of “discovered” and “robust”, respectively.

Line 32: Please write “had” instead of “have” here.

Line 82: “…this research is to…..” should be “…this research was to…..”.

Line 84-89: They are for materials and methods. Please remove them.

Line 91-96: I think you only collected Enterococcus isolates from the hospital which were previously isolated. Isn’t it? If yes, please start your materials and methods section by “Isolates selection”. You should mention all the information about the isolation of enterococci, e.g., how these organisms were isolated from how many and which type of samples. Then, go with your own work.

Also, is it important to mention the sex and age of the patients? Because you didn’t perform any risk factor analysis for the occurrence of enterococcal infections. If you want, you can omit them. Also, you can keep them.

Line 102: “Negative plates” means?

Line 105,107,108: “is” should be “was”.

Line 114: Please use “agents” instead of “drugs”.

Line 124: You mentioned MDR in your title. Please mention the definition of MDR with a reference here.

Line 116-118: Please mention the concentration of the antibiotics here. Also, please mention their classes here.

Another observation here. Why did you use streptomycin? You’ve already mentioned that enterococci are intrinsically resistant to aminoglycosides. Also, as far as I know, there is no zone diameter data for streptomycin against enterococci. Please clarify.

Subsections 2.4, 2.5, and 2.6: Please merge them as all of them have mentioned the procedure of biofilm formation. You can write 2.4.1. and 2.4.2. instead of 2.5 and 2.6., respectively. Subsequently, correct the number of other sub-sections.

Line 177: correlation or variation?

Line 179: Only chi-square test? I believe you had to perform a chi-square test for relatedness. If yes, please mention it.

Line 182: Please provide a reference for this statement.

Line 184-185: Please clarify the sentence. Are you providing your result here or just providing a statement? I believe it’s your result. If yes, please remove the reference.

Figure 1: I was just wondering didn’t you find any intermediate or moderate isolates?

Tables 1 and 2: Please mention the number of classes under every pattern. For MDR, the number of classes is more important than the number of antimicrobial agents.

Also, why 27? What about the other three isolates? If you want to keep them as only MDR, please rephrase the title of the table and mention the MDR isolates.

Line 236: a moderate adherent or moderate producers?

Figure 2: Please mention the meaning of black, green, or other colors.

Figure 4: What do you mean by P and NP? Please mention it here.

Figure 6: Why E. faecium DSM, NBRC, and E. faecalis NBRC here? Did you use them to compare your isolates with them? If yes, why only three? Why did you select only them? Please clarify it. And if possible, please mention them in your manuscript.

Line 264, 265, 266, 298: You have missed mentioning the appropriate sign (= or <). Please mention them here.

Line 317: Please remove “In conclusion”.

Line 317-319: Please rephrase the sentence.

Line 322: Chemicals?

Line 325: Supplementary information files? I didn’t find any supplementary materials.

Best wishes

Author Response

Kindly find the response to comments.

Round 2

Reviewer 2 Report

The authors addressed all of my comments. However, I have a few minor comments. Please check them below:

Line 81-82: Please remove this sentence.

Line 88: Please provide a reference for the protocol here.

Line 171: Please provide the name of the company, city, and country of gel electrophoresis. Also, please do the same for other products you used.

Line 174: Please provide the name of the company, city, and country of GraphPad Prism, i.e., (GraphPad Software, Inc., San Diego, CA, USA).

Table 1: Please write "No. of antimicrobial classes" instead of "No. of antibiotics belonging to different antimicrobial classes".

Line 248: Please write "reported" instead of "detected" here.

Line 401: Please write "found" instead of "detected" here.

Line 446: Please write "showed" instead of "detected" here.

I believe this manuscript can be published after addressing these minor comments.

Best wishes and congratulations.

Author Response

The authors addressed all of my comments. However, I have a few minor comments. Please check them below:

Line 81-82: Please remove this sentence.

Response: We appreciate the reviewer’s careful analysis of our study.This sentence has been removed.

Line 88: Please provide a reference for the protocol here.

Response: The reference has been provided: Declaration of Helsinki [22].

Line 171: Please provide the name of the company, city, and country of gel electrophoresis. Also, please do the same for other products you used.

Response: Gel electrophoresis (Cleaver Scientific, Warwickshire, UK), utilizing a 1.5 % agarose gel (Cleaver Scientific Ltd) was used to analyze the amplification products.

Line 174: Please provide the name of the company, city, and country of GraphPad Prism, i.e., (GraphPad Software, Inc., San Diego, CA, USA).

Response: All obtained data were analyzed by GraphPad Prism (GraphPad Software, Inc., San Diego, CA, USA)

Table 1: Please write "No. of antimicrobial classes" instead of "No. of antibiotics belonging to different antimicrobial classes".

Response: Corrected.

Line 248: Please write "reported" instead of "detected" here.

Response: Corrected.

Line 401: Please write "found" instead of "detected" here.

Response: Corrected.

Line 446: Please write "showed" instead of "detected" here.

Response: Corrected.

I believe this manuscript can be published after addressing these minor comments.

Best wishes and congratulations.
